# Sodium Tungstate (NaW) Decreases Reactive Oxygen Species (ROS) Production in Cells: New Cellular Antioxidant

**DOI:** 10.3390/biomedicines11020417

**Published:** 2023-01-31

**Authors:** Alejandro J. Yañez, Karen Jaramillo, Camila Blaña, Rafael A. Burgos, Adolfo Isla, Pamela Silva, Marcelo Aguilar

**Affiliations:** 1Facultad de Ciencias, Universidad Austral de Chile, Valdivia 5090000, Chile; 2Interdisciplinary Center for Aquaculture Research (INCAR), Universidad de Concepción, Concepción 4030000, Chile; 3Instituto de Farmacología y Morfofisiología, Facultad de Ciencias Veterinarias, UACH Campus Isla Teja, Valdivia 5090000, Chile; 4Departamento de Ciencias Básicas, Facultad de Ciencias, Universidad Santo Tomás, Valdivia 5090000, Chile

**Keywords:** diabetes, diabetic nephropathy, ROS

## Abstract

Diabetic nephropathy (DN) is the leading cause of end-stage renal failure worldwide. Hyperglycemia generates reactive oxygen species (ROS), contributing to diabetic complications, especially in DN. Sodium Tungstate (NaW) is an effective antidiabetic agent for short and long-term treatments of both type 1 and type 2 diabetes models. In this study, we evaluated the effect of NaW on ROS production in bovine neutrophils incubated with platelet-activating factor (PAF) and in HK-2 cells induced by high glucose or H_2_O_2_. In addition, we evaluated the effect on iNOS expression in the type 1 diabetic rat model induced with streptozotocin (STZ). NaW inhibited ROS production in PAF-induced bovine neutrophils, and human tubular cells (HK-2) were incubated in high glucose or H_2_O_2_. In addition, NaW inhibited iNOS expression in glomeruli and tubular cells in the type 1 diabetic rat. This study demonstrates a new role for NaW as an active antioxidant and its potential use in treating DN.

## 1. Introduction

Diabetes is a metabolic disease characterized by a continuous increase in blood glucose, affecting the cardiovascular, visual, renal, and nervous systems. Diabetic nephropathy (DN) occurs in a third of patients with type 1 DM (DM1) and type 2 (DM2) and is the leading cause of the end-stage renal disease (ESRD) worldwide [1].

Experimental evidence and clinical studies suggest a close relationship between hyperglycemia and oxidative stress [2]. The generation of reactive oxygen species (ROS) directly results from hyperglycemia and plays an important role in developing and progressing chronic diabetic complications, especially diabetic nephropathy [3,4,5,6]. ROS overproduction occurs in the kidney of patients with diabetes mellitus and diabetic nephropathy [7,8,9] and animal models of diabetic kidney disease [10]. Oxidative stress in the rat glomerulus is an early event in diabetes in vivo [11].

There is limited information on the possible role of neutrophils in DN. Neutrophils in patients with DM1 and DM2 are more able to adhere to the endothelium than in healthy patients. Neutrophils release large amounts of ROS, which promotes endothelial damage and thus accelerates DN-mediated kidney damage. Furthermore, an increased proportion of neutrophils is associated with albuminuria in diabetic patients [12]. Treatment with antioxidants, such as vitamin E or rotenone (an electron transport chain inhibitor), blocks ROS production and can reverse albuminuria, glomerular hypertension, and glomerular fibrosis [8]. Additionally, the over-expression of the enzyme superoxide dismutase reduces glomerular pathophysiological changes in diabetic rats induced with STZ, demonstrating that inhibiting oxidative stress can decrease the manifestations associated with DN [13,14,15]. These observations support the importance of ROS in the mechanism of kidney damage and support the use of antioxidant treatment as a potential therapeutic procedure for DN [8].

Nitric oxide synthase (NOS) has three main isoforms: inducible NOS (iNOS, type 1), neuronal NOS (nNOS, type 2), and endothelial NOS (eNOS, type 3). All isoforms are expressed in renal tissue. Both nNOS and eNOS are constitutively expressed, but iNOS is expressed in immune cells such as macrophages. The expression and activity of iNOS are regulated by signaling pathways, including the redox-sensitive transcription factor NF-κB. Increased glomerular filtration rate (GFR), renal plasma flow (RPF), increased FF filtration rate, and renal hypertrophy are physiological symbols of the early stages of diabetic nephropathy. In diabetic human and animal models, ultrafiltration often precedes microalbuminuria and histological changes indicative of diabetic nephropathy. In diabetic human and animal models, almost all forms of NOS increase with increased urinary excretion of NOS metabolites. Additionally, increased NOS can increase ROS and has been implicated in the pathogenesis of ultrafiltration [16].

Important inorganic compounds mimic the effect of insulin or increase its action when administered orally to diabetic rats. These include Vanadate, Selenate, Molybdate, Chromium, Cobalt, and Sodium tungstate [17]. Sodium tungstate (NaW) is an effective antidiabetic agent for short- and long-term treatments of both type 1 and type 2 diabetes models [18,19,20,21]. NaW treatment in STZ-induced diabetic rats normalizes glycemia by restoring hepatic glucose metabolism [19,20] and increases the total amount and GLUT4 translocation in muscle [21]. It regenerates pancreatic β-cells in rats treated with STZ [22], leads to weight loss through complex anti-obesity mechanisms, and uniquely does not cause hypoglycemic events. Other positive effects of NaW have been described in recent years, such as preservation of renal cell epithelial integrity, induction of chondrogenic differentiation of bone marrow mesenchymal stem cells for cartilage regeneration, visual enhancement of protective pathways, reduction of mechanical bone mass, and antiplatelet activity. NaW has received considerable attention in the search for new antidiabetic drugs over the past two decades because of its wide range of positive effects. However, neither its mechanism of action nor its side effects have been fully elucidated, although several preclinical studies have shown NaW is safe and non-toxic [23].

The effect of Sodium Tungstate on ROS production in DN is unknown. In this study, we evaluated the effect of NaW on ROS production in bovine neutrophils incubated with platelet-activating factor (PAF) and in HK-2 cells induced by high glucose or H_2_O_2_. In addition, we evaluated the effect on iNOS expression in the type 1 diabetic rat model induced with streptozotocin (STZ).

## 2. Materials and Methods

### 2.1. Diabetic Rat Model and Sodium Tungstate Treatment

We used male Sprague-Dawley (SD) rats weighing 200–250 g. For the induction of diabetes, rats were injected with 55 mg/kg of STZ intravenously (caudal vein), dissolved in 1 mL of autoclaved and filtered (0.2 µm) 10 mM sodium citrate, 0.9% NaCl, pH 4.5. 2 weeks after injection we confirmed the induction of diabetes. We split the animals into two groups. One group of diabetic animals was maintained for 4 months. The second group of diabetic animals was treated with NaW. In the second month post-diabetes induction, the rats were given a solution of 0.7 g/L of NaW in water ad libitum for two weeks. The dose was increased to 2 g/from the third week until the fourth month. In parallel, there was a control group without treatment and a control group treated with NaW, previously injected only with the vehicle (sodium citrate) (Figure 1).

### 2.2. Isolation of Bovine Neutrophils

Blood was obtained aseptically by jugular venipuncture in the tubes of ACD (citrate and dextrose). Subsequently, the tubes were gently shaken for 5 min. They were centrifuged at 1000× *g* at 20 °C for 20 min. The plasma and phlogistic layer were removed, while the remaining red blood cells and neutrophils were resuspended in cold HBSS (5.33 mM KCl, 0.441 mM KH_2_PO_4_, 4.17 mM NaHCO_3_, 137.93 mM NaCl, 0.338 mM Na_2_HPO_4_, 50.56 mM dextrose). It was centrifuged a second time at 1000× *g* at 20 °C for 20 min, and the supernatant was removed. Red blood cells were discarded by rapid lysis with cold phosphate buffer in an aqueous solution (5.5 mM NaH_2_PO_4_, 8.4 mM HK_2_PO_4_, pH 7.2) in a 2:1 ratio. Isotonicity was restored with hypertonic phosphate buffer (5.5 18 mM NaH_2_PO_4_, 8.4 mM HK_2_PO_4_, 0.46 M NaCl, pH 7.2) in a 1:1 ratio, subsequently centrifuged at 600× *g* at 20 °C for 10 min. The neutrophil pellet was resuspended and washed with cold HBSS at least twice, centrifuging each time at 500× *g* for 10 min at 20 °C. Finally, neutrophils were counted using a Neubauer chamber. Purity was determined using flow cytometry, and viability using the trypan blue exclusion method.

### 2.3. Estimation of Reactive Oxygen Species in Bovine Neutrophils Using Luminol

Neutrophils were stimulated with platelet-activating factor (PAF), and ROS generation was measured with a luminol assay (0.5 mM) in a luminometer (FLUOstart luminometer).

### 2.4. HK-2 Cell Culture and Treatments

Human HK-2 proximal tubule cells (ATCC) were cultured in K-SFM medium (GIBCO, Invitrogen, Carlsbad, CA, USA) containing 5% fetal bovine serum and 1% penicillin/streptomycin. Subsequently, the cells were exposed to different conditions; normal levels of D-glucose (5 mM), high D-glucose, or high L-glucose (25 mmol/L) with or without treatment with 1 mM Sodium Tungstate for 48 h.

### 2.5. Estimation of ROS Production in HK-2 Cells by Flow Cytometry

Cells were cultured with 5 mM or 30 mM glucose in the presence or absence of NaW or pre-treated for 30 min with 10 nM diphenyleneiodonium (DPI), an NADPH oxidase inhibitor. Subsequently, the cells were incubated with the fluorescent probe dichlorofluorescein (DCF) for 15 min at 37 °C, and fluorescence intensity was measured by flow cytometry. As a positive control, the cells were incubated for 15 min with phorbol-myristate-acetate (PMA).

### 2.6. Confocal Microscopy

100,000 HK-2 cells were seeded. Fluorescent incorporation in the cells was examined using a Fluoview 1000 confocal microscope. Cells were pre-incubated as described above for assays of the effect of both 0.1 mM NaW and 500 μM H_2_O_2_. They were incubated with 10 μM of DCFH-DA Probe for 30 min and 15 min with 100 nM MitoTracker (cell signaling) probe. After incubation with the probe, the excess was removed by washing three times with 1× PBS, and cells were visualized. The DCFH-DA probe was excited at 480 nm and emission at 530 nm, and the MitoTracker probe was excited at 644 nm and emission at 665 nm. A 60X objective, NA 1.49, oil immersion, Olympus was used.

### 2.7. Immunohistochemistry in Tissue

Kidney tissue samples were deparaffinized in xylene and rehydrated with degraded ethanol. Endogenous peroxidase activity was inhibited with 3% H_2_O_2_ for 5 min, and the tissue was blocked with 3% BSA in PBS and permeabilized with 0.3% Triton X-100 for 30 min. Primary antibody (Anti-iNOS Santa Cruz antibody ab283655) was incubated for 1 h and washed in PBS. The secondary antibody (Universal ICQ LSAB plus kit, DAKO Corporation, Carpenteria, CA, USA) was incubated for 20 min, followed by washing in PBS. The reaction was visualized with DAB, and hematoxylin was used for nuclear counterstaining.

### 2.8. Statistical Analysis

Values expressed as the mean ± standard error of the total number of experiments. Multiple statistical analysis *t*-test with Bonferroni correction was used. A value of *p* < 0.05 was considered statistically significant.

## 3. Results

### 3.1. The Effect of Sodium Tungstate on ROS Production in Bovine Neutrophils Stimulated with PAF

We first tested the effect of NaW on ROS in bovine neutrophils induced by PAF as a proof of concept. Subsequently, we analyzed the effect of NaW on ROS in tubular cells induced in high glucose and a type 1 diabetic mouse model.

The effect of NaW on ROS inhibition was tested using a recognized activator of respiratory burst in neutrophils, PAF. Bovine neutrophils stimulated with PAF generated higher ROS levels than control cells. ROS generation was inhibited when cells were coincubated with PAF and increasing concentrations of NaW (Figure 2).

### 3.2. The Effect of Sodium Tungstate on ROS Production in HK-2 Cells Incubated with High Glucose

HK-2 cells cultured for 4 h with high glucose (30 mM) generated a higher amount of ROS than control cells cultured in 5 mM glucose. Cells incubated in high glucose (30 mM) and treated with (10 μM, 100 μM, and 1 mM) NaW showed a significant reduction in ROS production compared to cells incubated with high glucose and cells treated with a ROS inducer, PMA. There was no ROS production in cells incubated with 5 mM glucose and 1 mM NaW (Figure 3).

Visualization of the effect of Sodium Tungstate on ROS production and mitochondrial activity in HK-2 cells.

There was no mitochondrial ROS production in control cells incubated in low glucose (5.5 mM glucose) or NaW. H_2_O_2_ treatment increased mitochondrial ROS, whereas NaW and H_2_O_2_ treatment decreased mitochondrial ROS production (Figure 4).

### 3.3. Effect of Sodium Tungstate on Inducible Nitric Oxide Synthase (iNOS) Expression in Diabetic Rats

The expression of iNOS increased in the glomeruli and tubular cells of diabetic animals compared with control animals. While in diabetic rats treated with NaW showed a decrease in iNOS expression in glomeruli and tubular cells compared to diabetic rats (Figure 5). 

## 4. Discussion

Maintaining a hyperglycemic state leads to the activation of oxidative stress, which is associated with diabetic kidney damage [24].

NaW treatment in STZ-induced diabetic rats normalizes glycemia by restoring hepatic glucose metabolism [18,19] and increases GLUT4 levels and translocation in the muscle [20]. Furthermore, NaW induces the regeneration of pancreatic β-cells in STZ-treated rats [22]. However, the effect of Sodium Tungstate on ROS production induced by hyperglycemia is unknown.

In cells cultured with high glucose (30 mM) or stimulated with PMA, treatment with 1 mM NaW reduced ROS production after 4 h of coincubation with high glucose. No changes were observed in cells incubated with 5 mM glucose and NaW compared to cells treated with 5 mM glucose without NaW. ROS production in bovine neutrophils was studied to corroborate the effect observed in HK-2 cells. In this cell type, NaW inhibited the respiratory burst of cells stimulated with PAF in a matter of seconds, indicating a rapid effect of NaW on the inhibition of ROS production. Cells incubated in low glucose (5.5 mM glucose) or NaW had no ROS or mitochondrial activity. Cells treated with H_2_O_2_ generated an increase in ROS, while cells treated with NaW and H_2_O_2_ generated a decrease in ROS and mitochondrial activity. The expression of iNOS increased in the glomeruli and tubular cells of diabetic animals compared with control animals. While in diabetic rats treated with NaW showed a decrease of iNOS in glomeruli and tubular cells compared to diabetic rats.

Polymorphonuclear neutrophil leukocytes (PMNs) are an important effector cell in human DN. Human PMNs can release ROS and induce tissue damage. ROS such as superoxide, H_2_O_2_, hydroxyl radical, and the myeloperoxidase-H_2_O_2_-halide system can induce tissue damage. Evidence has suggested that H_2_O_2_ may cause the highest levels of kidney damage of all the ROS species released by PMNs [25].

PMNs release large amounts of ROS, which promote endothelial damage and thus accelerate DN-mediated kidney damage. Furthermore, increased neutrophil levels are associated with albuminuria in diabetic patients [12]. Multiple studies relate the adverse effects observed in diabetes as a consequence of increased ROS production at the renal level in high glucose environments, acting as a reciprocal inducer and amplifier of the cell signaling events that occur in these conditions [7,8,9]. ROS production is considered a central factor in the generation of kidney damage since it can directly damage the molecules and act as a secondary messenger. In addition, ROS can activate pro-inflammatory and pro-fibrotic pathways [26,27,28], such as nuclear factor NF-κB [29], and favor renal fibrosis by increasing TGFβ1 expression [30]. In turn, this can increase ROS production, resulting in a vicious circle, eventually resulting in the pathological accumulation of extracellular matrix proteins and renal fibrosis, and impacts the renal damage induced by hyperglycemia [31].

Studies have shown increased iNOS in renal tissue in streptozotocin (STZ)-induced diabetic rats and in tubular cells (HK-2) treated with high glucose [6,32]. The treatment of NOS inhibitors ameliorates hyperfiltration in diabetes. Additionally, increased NADPH diaphorase staining was found in afferent arterioles, reflecting constitutive NOS activity and suggesting that increased NO production may be associated with ultrafiltration (assessed by creatinine clearance) in diabetic rats. These changes normalized after treatment with NOS inhibitors such as L-NAME [33]. 

## 5. Conclusions

NaW inhibited ROS production in PAF-induced bovine neutrophils and in human tubular cells (HK-2) incubated in high glucose or H_2_O_2_. In addition, NaW inhibited iNOS expression in glomeruli and tubular cells in the type 1 diabetic rat. This study demonstrates a new role for NaW as an active antioxidant and its potential use for the treatment of DN.

## Figures and Tables

**Figure 1 biomedicines-11-00417-f001:**
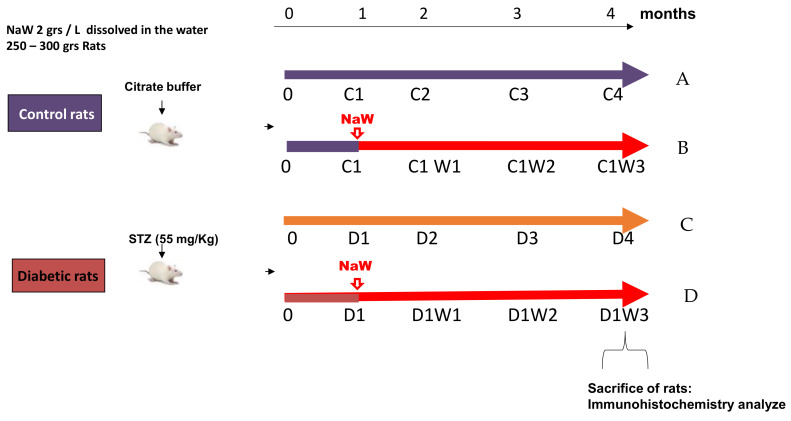
Scheme of treatment with Sodium Tungstate in control and diabetic rats. (**A**) Healthy control rats (C4), (**B**) NaW-treated control rats (C1W3), (**C**) diabetic rats (D4), and (**D**) NaW-treated diabetic rats (D1W3).

**Figure 2 biomedicines-11-00417-f002:**
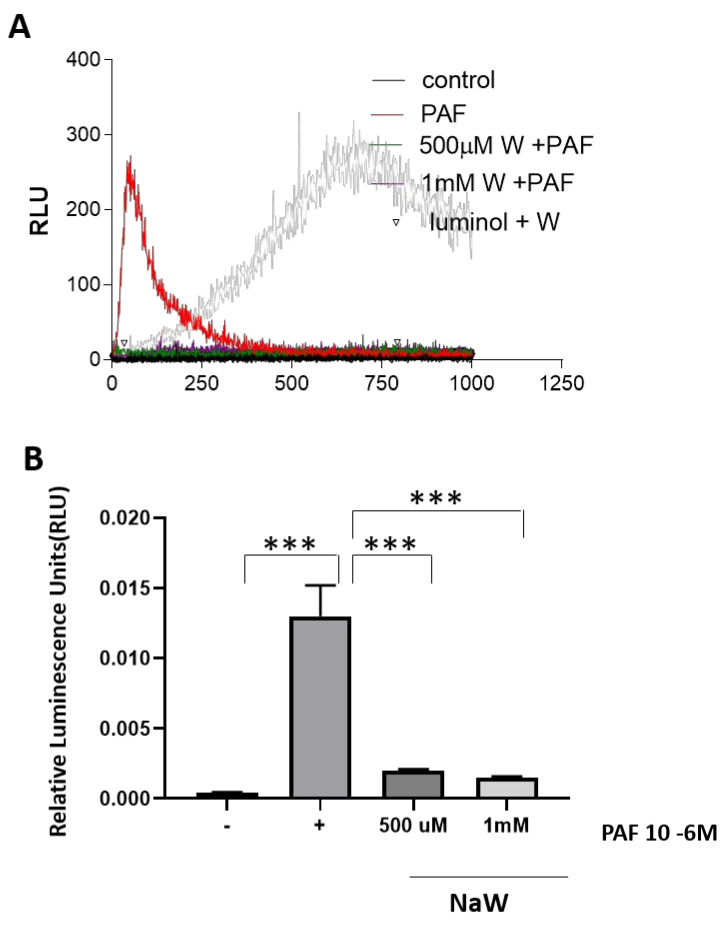
Effect of Sodium Tungstate on ROS production in bovine neutrophils stimulated with PAF. (**A**) Kinetics of ROS generation in neutrophils under basal conditions stimulated with PAF and treated with different concentrations of NaW. (**B**) Quantifying ROS generation in neutrophils under basal conditions stimulated with PAF and treated with different concentrations of NaW (500 µM y 1 mM). Cells were stimulated with 100 nM PAF, and ROS generation was measured using luminol (0.5 mM). Values reported as the mean ± standard error of 3 separate measurements. *** *p* < 0.0001 compared to the control (−PAF). *** *p* < 0.0001 compared to (+PAF), using a Multiple *t*-test, followed by Bonferroni correction for multiple comparisons.

**Figure 3 biomedicines-11-00417-f003:**
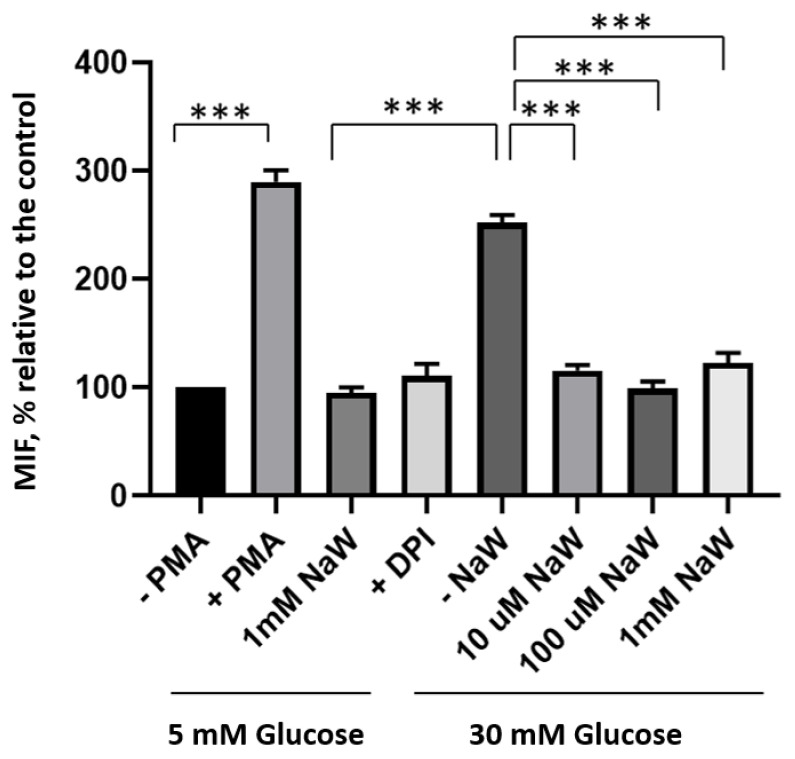
Effect of Sodium Tungstate on ROS production in HK-2 cells incubated with high glucose. ROS generation in HK-2 cells cultured with 5 mM glucose or 30 mM glucose for 4 h, with different concentrations of NaW (10 μM, 100 μM and 1 mM). Cells were stimulated with PMA for 15 min as a positive control. DPI was used as a negative control, pre-incubating the cells for 30 min prior to treatment with glucose and NaW. ROS production was measured by flow cytometry using the DCF fluorescent probe. Values reported as the mean ± standard error of 3 separate measurements. *** *p* < 0.0001 compared to the control (−PMA). *** *p* < 0.0001 compared to NaW. *** *p* < 0.0001 compared to control (−NaW), according to the Multiple *t*-test, followed by Bonferroni correction for multiple comparisons.

**Figure 4 biomedicines-11-00417-f004:**
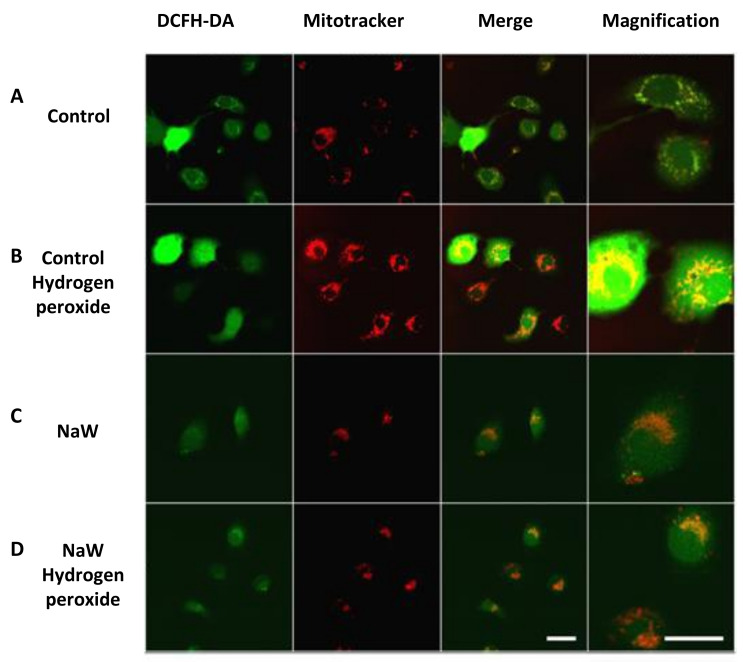
Effect of Sodium Tungstate at the mitochondrial level on ROS production in HK-2 cells cultured with H_2_O_2_. In vivo fluorescence for ROS observation in HK-2 cells cultured in low glucose 5.5 mM glucose at mitochondrial level (red) and cytosolic ROS (green): (**A**) Control cells. (**B**) Induced with 500 µM H_2_O_2_. (**C**) Cells treated with NaW. (**D**) Cells treated with NaW induced with H_2_O_2_. The probe used for ROS measurement is DCFH-DA 10 μM incubated for 30 min, while for mitochondria labeling, used MitoTracker 100 nM probe was incubated for 15 min. The magnification bar indicates 10 μm.

**Figure 5 biomedicines-11-00417-f005:**
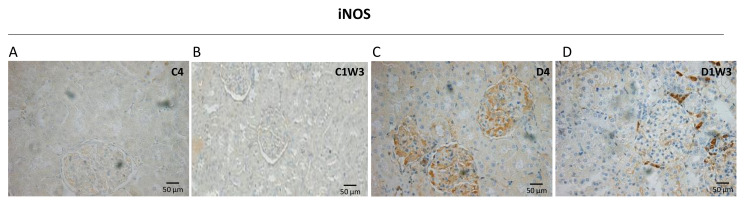
Effect of Sodium Tungstate on iNOS expression in diabetic rats. Location iNOS in glomeruli and tubular cells in the cortex. (**A**) healthy control rats (C4), (**B**) NaW-treated control rats (C1W3), (**C**) diabetic rats (D4), and (**D**) NaW-treated diabetic rats (D1W3). The values represent the mean ± standard error of the total number of animals per group. The scale bar indicates 50 µm.

## Data Availability

Not applicable.

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
