# Peer review of "Sodium Tungstate (NaW) Decreases Reactive Oxygen Species (ROS) Production in Cells: New Cellular Antioxidant"

_biomedicines, 2023, doi:10.3390/biomedicines11020417_

Round 1

Reviewer 1 Report

It is a very valuable finding on Sodium Tungstate (NaW) on ROS production in bovine neutrophils stimulated with platelet-activating factor (PAF) and in HK-2 cells stimulated with high glucose or hydrogen peroxide. The manuscript is well thought out and written; however, some minor comments need to be answered. 

1. The authors should justify the following:

(a) How the dosage fixed?

(b) Why Acute toxicity study was not conducted?

(c) Is this study was approved in institutiona ethical commitees? If Yes, then ethical number should be mentioned.  

(d) Why not positive control group taken?

 2. There are some minor grammatical and syntax errors that should be corrected.

Author Response

We appreciate the reviewer’s positive evaluation of our work and constructive comments. The comments below are all valuable and very helpful for revising and improving our manuscript. We have studied comments carefully and have made correction which we hope meet with approval. Next, I will answer your questions in detail below one by one. All the changes have been highlighted in the revised manuscript. I would like to thank you very much for your recognition and hard work on this manuscript again.

Reviewer 2 Report

Yañez A. et al., showed the administration of Sodium Tungstate (NaW) might inhibit ROS production in cell and is new therapeutic agent to treat Diabetic nephropathy (DN) looks interesting.

Although, this is an interesting manuscript, however author need to address lots of work to improve their study. Some of the points are listed below

1.     Elaborate more about Sodium tungstate (NaW) in introductory part.

2.     Please cite Oxidative stress in vivo is an early event of diabetes in the rat glomerulus, written in introduction part on page 1.

3.     What author means ROS overproduction written in introduction part? Is there any difference between increased ROS and ROS overproduction?

4.     Fig. 2 legend, what is the meaning of  “W” in 500uM W+ PAF, 1mM W+PAF

5.     In materials and method section, I wonder why the dose of NAW was changed from, 0.7g/L to 2g/L ?

6.     In Result 3, did author mean to say induction of mitochondrial activity increase ROS? How  author elaborate induction of mitochondrial activity increased ROS and is more vulnerable to Diabetic nephropathy (DN)?

7.     In Fig. 3, ROS level was not increased in low glucose, however in fig 4, author showed the increased ROS level in low glucose (5.5mM), how is it possible?

8.     Also in Fig. 4 legend, what does scale bar mean? Is it for lower magnification or higher magnification?

9.     Author should check the mRNA level of iNOS to validate their IHC

Author Response

(The authors gave the same response as above.)

Round 2

Reviewer 2 Report

Authors satisfied almost my queries